# On the Nature of Axiverse-Axion Cosmic Strings

John March-Russell[1] and Hannah Tillim[1,2]

**1** Rudolf Peierls Centre for Theoretical Physics, University of Oxford, Beecroft Building, Oxford OX1 3PU, United Kingdom
**2** Department of Physics and Astronomy, Johns Hopkins University, Baltimore, MD 21218, USA

January 25, 2023

## Abstract

**The string axiverse of axion-like-particles (ALPs) can comprise the QCD axion, dark matter, and quintessence. It has been assumed without proof that, like field-theory ALPs, such string-theory ALPs lead to early universe cosmic strings. We study such axiverse cosmic string solutions, including moduli stabilization, finding the string cores explore moduli space boundaries with central wrapped branes. Equivalence principle violating variations in Standard Model parameters are also possible.**

## 1   Introduction

One of the most compelling motivations for physics beyond the Standard Model is the strong-CP problem [1]. The Peccei-Quinn (PQ) solution [2] employs a new light pseudo Nambu-Goldstone boson (pNGB) [3,4], the QCD axion, $a(x)$, coupling to QCD via an anomaly term. Non-perturbatively QCD generates a periodic potential for $a$, giving it a mass, and the QCD $\bar{\theta}$-parameter automatically relaxes to zero. In addition to solving the strong CP-problem, the axion is also a successful cold dark matter (DM) candidate [5–7] with a rich phenomenology.

For the PQ-mechanism to work, however, the QCD potential must dominate all other sources of explicit PQ breaking by $\gtrsim 10^{10}$ [8–10], a non-trivial constraint as there is strong

evidence that all global symmetries are explicitly broken by gravity [11–16]. Favouring the PQ solution is the fact that (possibly heavy) axion-like-particles (ALPs) are ubiquitous in string theory [17–19]. In fact, keeping the QCD axion sufficiently light in string theory implies there exist *many light ALPs* $a^i(x)$ [19] - the string axiverse - one of which is the QCD axion while others could be DM or quintessence. Unsurprisingly axiverse ALPs are thus the focus of an extremely active experimental and observational detection effort.

Importantly for our purposes ALPs are always compact fields $a^i(x) \in [0, 2\pi f^i]$ where $f^i$ is an energy scale. Thus in addition to ALP particle excitations one may also consider *topologically non-trivial axion cosmic string solitons* where $a^i(x)$ winds $a^i \to a^i + 2\pi n f^i$ $(n \in Z)$ as some loop in physical space is traversed [20, 21]. In fact in traditional QCD axion models [22–25] $a(x)$ arises as the phase of the field whose vacuum expectation value (VEV) spontaneously breaks global U(1) PQ symmetry, $\Phi(x) = |\Phi(x)| \exp(ia(x)/f)$, with VEV $\langle|\Phi(x)|\rangle = f/\sqrt{2}$. Then, for a cosmic string solution, $|\Phi^i(x)|$ necessarily has a zero at some location, and this string core region explores and is sensitive to UV physics. Although such axion strings become the boundaries of axion domain walls once the ALPs acquire a mass, the decaying string/domain-wall network is important for both relic axion DM production [26–30] and the generation of stochastic gravitational wave backgrounds [21, 31–34].

The origin and UV physics of axiverse ALPs is quite different to traditional QFT ALPs however (see Sec. 2), so one anticipates that axiverse cosmic string solutions, if they exist, will be unlike those of QFT axions, at least in their cores. While some phenomenological aspects of axiverse cosmic strings have been previously discussed there has been no proper analysis of the solutions themselves in this important case, leaving their status and associated physics uncertain. This is the subject of our work

## 2   String Axiverse Recap

Axiverse ALPs arise from the $k$-form fields present in the underlying string theory, such as the 2-form, $B_2$, of heterotic and type II, or the type II RR $k$-forms $C_k$ (see Sec.5). Vitally, a *single* $k > 1$ form leads to a *multiplicity* of classically massless ALP candidates, $a^i$, determined by the topology of the 6d Calabi-Yau (CY) compactification, $Z$. For example, if $\omega_i$ is a basis for the $h^{1,1}$ complex (1,1) harmonic forms (dual to closed 2-cycles) of $Z$ then $B_2$ gives $h^{1,1}$ potential 4d ALPs via $B_2 = b^i(x)\omega_i$. Related arguments apply to other $k$-form fields and $k$-cycles.

Crucially, as realistic compactifications are topologically rich, with $10^{\text{few}}$ cycles, there are a large number of potentially light 4d ALPs. A subset are lifted at tree level, but those that survive remain massless in perturbation theory. The most attractive compactifications also preserve $\mathcal{N} = 1$ supersymmetry (SUSY) before moduli stabilization, so the $a^i$ are accompanied by 'saxions' $t^i$ - scalar compactification moduli. The leading 4d effective action of the moduli fields $m^i \equiv a^i(x) + i t^i(x)$ before including the stabilization potential is $(m^{\bar{i}} = (m^i)^*)$

$$\mathcal{L} = G_{i\bar{j}}(m, \overline{m}) g^{\mu\nu} \partial_\mu m^i \partial_\nu m^{\bar{j}} \tag{1}$$

where $G_{i\bar{j}} \equiv \partial_i \partial_{\bar{j}} K(m, \overline{m})$ with $K$ the Kähler potential. The behaviour of $G_{i\bar{j}}$ at moduli space boundaries will be of special interest for the construction of axiverse strings.

A tree level vanishing moduli potential is protected from loop corrections but generated by non-perturbative effects, which, as seen in realistic stabilization mechanisms, involve SUSY-breaking. The $t^i$ receive masses, while the $a^i$, being pNGBs, can be separately protected and often receive *hierarchically smaller* non-perturbative masses. In fact, this hierarchy in masses essentially must exist if axiverse ALPs are light enough to be relevant for the strong CP-problem or light 'field' DM.

# 3 Axiverse Strings: Basics

Consider the simple case of compactification on a 6-manifold $X_4 \times T^2$ with ALP arising from the 2-form $B_2$ with legs in the $T^2$ directions, $y_m$, $m = 1, 2$ (we do not need to specify $X_4$). If $h_{mn}$ is the $T^2$ internal metric, then $\rho(x) \equiv B_{12} + i\sqrt{\det h} \equiv \rho_1 + i\rho_2$ is the dimensionless complexified Kähler 'size' modulus, and $\rho_1$ is our ALP of period 1. The domain, $\mathcal{F}$, of inequivalent $\rho$ fields is the upper-half $\mathbb{C}$-plane modulo $SL(2, \mathbb{Z})$ - see Fig.1. The reduction of the 10d string action gives $K(\rho, \overline{\rho}) \propto \log(\rho_2)$, so $G_{\rho\overline{\rho}} \propto 1/4\rho_2^2$, and the relevant Einstein-frame 4d effective action of the light fields is

$$S = M^2 \int d^4x \sqrt{-\det g} \left\{ g_{\mu\nu} \frac{\partial^\mu \rho \partial^\nu \overline{\rho}}{4\rho_2^2} - V(\rho_2) \right\} . \tag{2}$$

Here the 4d dilaton $\varphi$ does not appear as we assumed it to be stiffly stabilised. (In Sec.4. we return to this and discuss the physics that sets the scale $M$.) In Sec.6 the effect of an additional non-perturbative potential for the axion is discussed assuming a well-separated hierarchy of scales $V(\rho_2) \gg \tilde{V}(\rho_1)$, so $\rho_2$ is much heavier than $\rho_1$.

A crucial feature of eq.(2) is that the axion decay constant is set by the partner VEV: $f_{\text{eff}} = M/(\sqrt{2} \langle \rho_2 \rangle)$. Thus, by analogy with usual QFT axion strings where $f_{\text{eff}}(x) = \sqrt{2} \langle |\Phi(x)| \rangle$ has a zero at the string core, we expect *axiverse cosmic strings to have cores where $\rho_2 \to \infty$, namely at a suitable boundary of the moduli space*. This expectation is essentially correct, and generalises to axiverse strings arising from other compactifications.

Concerning $V(\rho_2)$, it must go to zero in the decompactification limit [35–39], $\rho_2 \to \infty$, and, for stabilization, have a local minimum at some $\rho_2 = b > 0$, with $V(b) \simeq 0$. We require $b \gg 1$ so that the stabilizing $T^2$ is large and the leading-order action, eq.(1), is appropriate. This is also necessary to sufficiently suppress the non-perturbative effects that would otherwise lift all ALP masses and eliminate the axiverse [19]. We thus take $V(\rho_2)$ to be of the form shown in Fig.2.

A simple potential with these properties is

$$V(\rho_2) = \mu^2 e^{-a\rho_2}(\rho_2 - b)^2 , \tag{3}$$

where $\mu^2 e^{-ab} \ll M^2$ sets the scale. The canonical modulus mass is $m_c = 2b\mu e^{-ab/2}$ at $\langle \rho_2 \rangle = b$. The solution is indifferent to the shape of $V$ for $\rho_2 < b$ as long as $b$ is a minimum, while the exponential decline of $V(\rho_2 \to \infty)$ in eq.(3) is for convenience - all results are qualitatively unaffected as long as $V(\rho_2 \to \infty) \sim 1/\rho_2^n$ for $n > 0$.

To proceed initially freeze the metric $g_{\mu\nu}(x) = \eta_{\mu\nu}$. For an infinite string along the $x_3$-axis the solution depends only on the transverse coordinates $z \equiv x_1 + ix_2 \equiv r \exp(i\theta)$. As analysed in Ref. [40] in the absence of $V$, the equation following from eq.(2) is (here $\partial = \partial/\partial_z$ etc)

$$\partial \overline{\partial} \rho - \frac{2\partial \rho \overline{\partial} \rho}{\rho - \overline{\rho}} = 0 . \tag{4}$$

which is solved by any (anti)meromorphic $\rho$. However, these solutions are not directly physically relevant as they are greatly altered by the potential. Thus consider the ansatz

$$\begin{aligned} \rho_1 &= \theta/2\pi, \\ \rho_2 &= \rho_2(r) \end{aligned} \tag{5}$$

for the elementary winding number $w = 1$ string. This leads to

$$\frac{d^2\rho_2}{dr^2} + \frac{1}{r}\frac{d\rho_2}{dr} - \frac{1}{\rho_2}\left(\frac{d\rho_2}{dr}\right)^2 + \frac{1}{4\pi^2 r^2 \rho_2} = 2\rho_2^2 \frac{dV}{d\rho_2} , \tag{6}$$

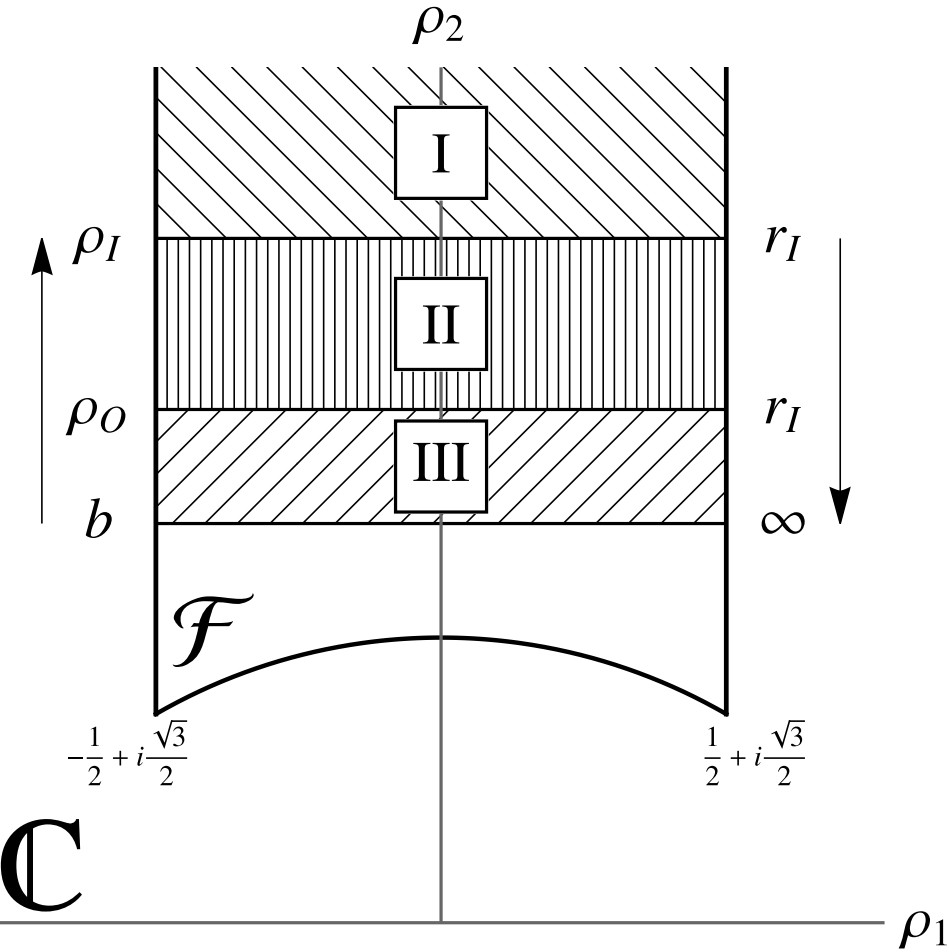

Figure 1: Domain $\mathcal{F}$ of $T^2$ Kähler moduli space with regions explored by I) inner core $(r < r_I; \rho_2 > \rho_I)$, II) outer core $(r_I \leq r \leq r_O)$, III) far field $(r > r_O; b \leq \rho_2 < \rho_O)$, indicated.

with $\rho_2(r \to 0) \to \infty$, and $\rho_2(r \to \infty) \to b$. The solution is then found by stitching together the three regions - see Figs..1,2:

Region I) Inner core: $0 < r < r_I, \infty > \rho_2 > \rho_I$

Region II) Outer core: $r_I \leq r \leq r_O, \rho_I \geq \rho_2 \geq \rho_O$

Region III) Far field: $r_O < r < \infty, \rho_O > \rho_2 > b$

For $V(\rho_2)$ of eq.(3) we take matching values $\rho_O \equiv b + 1/a$ and $\rho_I \equiv b + 8/a \gg 1$. The features of the string solution do not depend on these specific choices.

In Region III $V$ is important, and the solution asymptotes to $\langle \rho_2 \rangle = b$. In the inner-core, where $\rho_2 \to \infty$, $V$ can be neglected. In between there is an 'outer core' transition region, either 'thick' or 'thin'. In more detail, the $w = 1$ solution in the $r > r_O$ region is

$$\rho_2 = b + \frac{b}{(2\pi b \, m_c r)^2} + \dots , \tag{7}$$

with $1/r^2$ behaviour characteristic of global cosmic strings. Given $\rho_O = b + 1/a$ the outer core

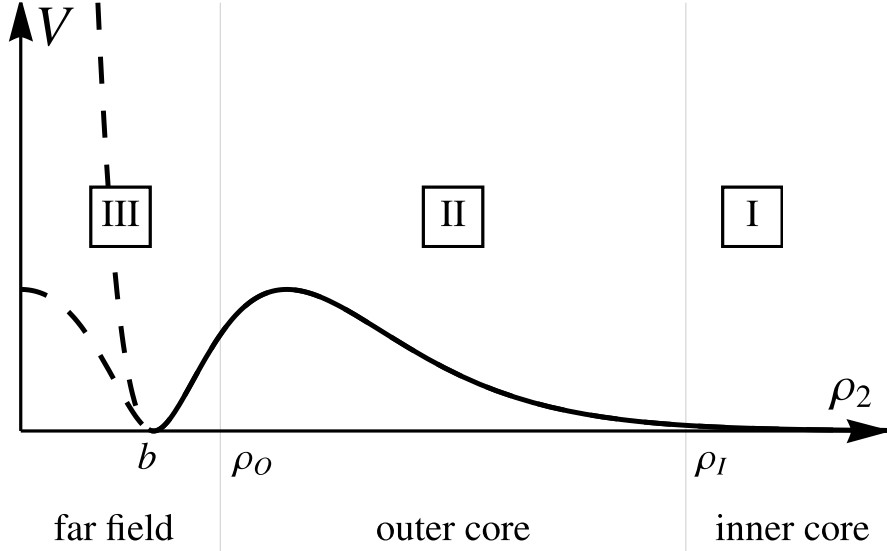

Figure 2: General shape of $V(\rho_2)$, with three regions delineated. Solid/dashed potential curves illustrate the insensitivity of the cosmic string solution to the form of $V$ for $\rho_2 < b$.

radius is then $r_O \simeq (a/b)^{1/2}/2\pi m_c$. In the inner core the RHS of eq.(6) may be neglected, giving an approximate $w = 1$ solution[1] valid for $\rho > \rho_I = b + 8/a \gg 1$,

$$\rho(z,\bar{z}) \simeq -\frac{i}{2\pi}\log(cz), \quad c = \frac{\exp(-2\pi\rho_I)}{r_I}\ . \tag{8}$$

Finally the non-universal, numerically-derived, outer-core solution stitches Region I and III together.

The action, eq.(2), implies the $w = 1$ tension is

$$T_1 = 2\pi M^2 \int dr\, r\left[\frac{(\partial_r\rho_2)^2}{\rho_2^2} + \frac{1}{4\pi^2 r^2 \rho_2^2} + \mu^2 e^{-a\rho_2}(\rho_2 - b)^2\right] \tag{9}$$

Splitting this into contributions $T_{I,II,III}$ from the three regions, analysis shows that the far-field part $T_{III}$ is log-sensitive to the IR cutoff, $L$, while the leading inner-core contribution gives $\Delta T_I = 2M^2/\rho_I$, independent of $r_I$ and *finite* despite the decompactification. Minimising $T_1$ with respect to $r_I$ for fixed $L, r_O$ shows that the thin wall case requires $a \gg 14\pi$. For both thin and thick wall cases, the tension at distance $r > r_O$ is logarithmic IR sensitive

$$T_1(r) \simeq \frac{M^2}{2\pi b^2}\log\left(\frac{r}{r_O}\right) + \mathcal{O}\left(\frac{2M^2}{b}\right), \tag{10}$$

similar to a normal global cosmic string.

Finally note that the action eq.(2) and domain of the $T^2$ complex structure (CS) modulus $\tau$ are identical to that for $\rho$, so these results apply to $T^2$ CS strings too.

---

[1]As the $\rho$ domain is $\mathcal{F}$, rather than a strip of $\mathbb{C}$ as naively given by eq.(8), a more precise core solution is $j(\rho) = 1/cz$ where Klein's modular $j$-function maps the fundamental domain, $\mathcal{F}$, to $\mathbb{C}$ [40]. But for *realistic* strings with moduli stabilised this is only needed for $c|z| \ll 1$. In terms of $q \equiv \exp(2\pi i\rho)$, $j(\rho) \simeq q^{-1} + 744 + \mathcal{O}(q)$ for $|q| \ll 1$, so $\rho(z) \simeq -i(\log cz + 744\,cz + \ldots)/2\pi$ agreeing with eq.(8) up to errors of $< 1\%$ for $\rho_2 > b \geq 2$.

# 4 Gravity and The Core

For $r > r_O$ the tension of an axiverse string goes as a conventional global string, naively IR-divergent, eq.(10). For an isolated infinitely long static string this leads to a metric singularity at large distances [41], but for the realistic situation of a string network the inter-string separation gives a finite effective $L$, and the metric is smooth. At intermediate distances, $r_O < r < L$, and in the observationally-required limit $k(r) \equiv 4G_N T_1(r) \ll 1$, the 4d metric is well approximated by $ds^2 \simeq (1 - k(r) + k_0)(dt^2 - dx_3^2) - dr^2 - (1 - 2k(r))r^2 d\theta^2$ implying both an $r$-dependent deficit angle $\Delta\theta \simeq 2\pi k(r)$ and the well-known gravitational repulsion [42]. The $V(\rho_2)$-dependent constant $k_0$ is determined by matching to the core solution.

The mass scale, $M$, of the action, eq.(2) - equivalently the axion decay constant - may be greatly reduced compared to $M_{\text{pl}}$ if warped compactifications [18, 43, 44] are considered with the cycle on which the ALP is defined localised in the IR region with warp factor $e^{-w_0} \ll 1$. $M$ may also be lowered in SM-on-a-brane large volume compactifications with large string length $\ell_s \equiv 2\pi\sqrt{\alpha'}$ [18]. Both constructions allow $4G_N T_1(r) \ll 1$ consistent with the gravitational constraints on cosmic strings [45].

In the inner core the ansatz $ds_4^2 = dt^2 - dx_3^2 - p(z, \overline{z})dzd\overline{z}$ does not alter eq.(4), so eq.(8) is still a solution [40]. Analysis of the 4d Einstein equations shows that $p(z, \overline{z}) \sim -\log(r)$ as $r \to 0$. Importantly, since the decompactification is localised in 4d, $M_{\text{pl}}$ remains *finite*, set by the asymptotic $\langle\rho_2\rangle = b$. Strictly, though, the 4d effective theory breaks down for *local* experiments as $r \to 0$. A better description follows from a 6d theory with $\rho_1, \rho_2$ replaced by $B_2, h_{mn}$. One then sees that the $B_2$ and metric configuration is due to an effective string in 6d oriented along the $x^3$ axis that magnetically sources $B_2$ – the "solitonic" NS5-brane of heterotic or type II theories wrapped on the $X_4$ part of $X_4 \times T^2$. Such an effective string is well known [46, 47]. Integrating the NS5 $B_2$ flux over the boundary at $r \to \infty$ reproduces the integer winding number of the axiverse string.

The dilaton $\phi$ becomes dynamical at scales $r \lesssim 1/m_\varphi$, and the physics deviates from the inner-core solution of Sec.3. The full 10d NS5-brane solution [46] becomes asymptotically operative as $r \lesssim \min(1/m_\varphi, R_4)$ where $R_4$ is the linear size of $X_4$. (In other cases, involving ALPs from RR $k$-forms the effective string at the core of the solution is a suitable wrapped D-brane.) The connection of such wrapped brane sources with axionic cosmic string solutions has also been recently noted [48, 49].

# 5 General Axiverse Strings

We can generalise this analysis to large classes of axiverse strings. The solution depends on the $G_{i\overline{j}}(m, \overline{m})$ metric behaviour in the limit $G_{i\overline{j}} \to 0$, as well as the potential $V(m, \overline{m})$. We assume $V$ has three properties: a) There is a local minimum with $V \simeq 0$ at which all non-ALP saxion moduli, $\text{Im}(m^i)$, are stabilised; b) stabilization occurs at $\langle\text{Im}(m^i)\rangle = b^i \gg 1$ to have control over the 4d effective action; c) For $\text{Im}(m^i) \to \infty$, $V \to 0$.

The string solution is then a semi-geodesic "force-modified-motion" on the moduli space found by solving

$$\nabla^2 m^i + \Gamma^i_{jk}(m)(\nabla m^j).(\nabla m^k) = \partial_i V(m) , \tag{11}$$

subject to appropriate boundary conditions. Here $\Gamma$ is the connection derived from $G$. The metric is most important in the core where the effects of $V$ are small.

In addition to the heterotic and type II ALPs arising from $B_2$ (Sec.2), type II strings on a CY orientifold $Z$ also have ALPs arising from RR $k$-forms $C_k$. As orientifolds now have fields and basis forms split into even/odd sectors, for IIA the result is $h^{1,1}_-$ complex Kähler moduli $T^i$ defined by expanding $B_2 + iJ = T^i\omega_i = (b^i + it^i)\omega_i$ where $\omega_i$ is a basis of $h^{1,1}_-$ odd (1,1)-forms

and $J$ is the Kähler (1,1)-form of $Z$. The saxion components $t^i$ measure the (dimensionless) size of 2-cycles of $Z$. These are accompanied by $h^{2,1}+1$ CS moduli, including the axio-dilaton, with ALPs $\beta^a$ defined by $C_3 = \beta^a(x)\alpha_a$. Here $\alpha_a$ are a basis of even harmonic 3-forms on $Z$ [18, 50].

For IIB, there are $h^{1,1}_+$ Kähler moduli with ALPs $c^i$ defined by $C_4 = c^i(x)\tilde{\omega}_i$ where $\tilde{\omega}_i$ are a basis of even $(2,2)$ forms dual to the even $(1,1)$-forms. The associated saxion components $\tau^i(x)$ measure the size of 4-cycles of $Z$. In addition there are $h^{1,1}_-$ ALPs $d^i$ arising from $C_2 = d^i(x)\omega_i$, and also the 4d axio-dilaton field $S$ [50, 51].

A fraction of these ALPs can be made massive at tree level by a variety of mechanisms, but some remain massless before non-perturbative lifting, and are candidates for cosmic strings. (A wide variety of ALPs also exist in Type I string theory and M-theory with similar features.)

Turning to the kinetic metric, consider, eg heterotic and IIA Kähler $T^i = b^i + it^i$ moduli. Since for a 6d CY its volume $\mathcal{V}(x) = \kappa_{ijk}t^i t^j t^k/6$ where $\kappa_{ijk} = \int_Z \omega_i \wedge \omega_j \wedge \omega_k$ is the integer-valued triple intersection number, and the leading Kähler potential is of form $K = -2\log(\mathcal{V} + \dots)$ one finds for heterotic and IIA cases

$$G^{B_2}_{i\bar{j}}(x) = \frac{\kappa_{ikl}t^k t^l \kappa_{jmn}t^m t^n}{16\mathcal{V}^2} - \frac{\kappa_{ijk}t^k}{4\mathcal{V}} + \dots \tag{12}$$

up to corrections which are small if $t^i \gg 1$. Note that $G^{B_2}(t^i \to \infty) \sim 1/t^2$ just like the $1/\rho_2^2$ scaling of the $T^2$ metric. Up to a factor of the coupling $g_s^2$ the metric for the $C_2$ and 2-cycle $d^i$ ALPs in IIB is identical to eq.(12), while for the $c^i$-moduli of IIB arising from $C_4$ and 4-cycles

$$G^{C_4}_{i\bar{j}}(x) = g_s^2\left(\frac{t^i t^j}{8\mathcal{V}^2} - \frac{(\kappa_{ijk}t^k)^{-1}}{2\mathcal{V}}\right) + \dots, \tag{13}$$

an implicit function of the 4-cycle (strictly, divisor) volumes $\tau_i = \kappa_{ijk}t^j t^k/2$. As $\tau^i \to \infty$, $G^{C_4} \sim 1/\tau^2$.

A simple two Kähler moduli example is the IIB orientifold of a degree-18 hypersurface in $\mathbb{P}^4_{(1,1,1,6,9)}$ with $\mathcal{V} = (3t_1^2 t_5 + 18t_1 t_5^2 + 36t_5^3)/6$ where $t_1, t_5$ are the 2-cycle volumes. (For this model all CS/dilaton moduli can be flux stabilised [52], while Ref. [53] demonstrated successful Kähler moduli stabilization.) The 4-cycle volumes, the partners of the $c^{4,5}$ ALPs from $C_4$, are $\tau_4 = t_1^2/2$ and $\tau_5 = (t_1 + 6t_5)^2/2$ giving $\mathcal{V} = (\tau_5^{3/2} - \tau_4^{3/2})/9\sqrt{2}$. The $G^{C_4}$ metric is thus

$$\frac{g_s^2}{36\mathcal{V}^2}\begin{pmatrix} [2\tau_4 + \tau_5^{3/2}/\tau_4^{1/2}]/3 & -(\tau_4\tau_5)^{1/2} \\ -(\tau_4\tau_5)^{1/2} & [2\tau_5 + \tau_4^{3/2}/\tau_5^{1/2}]/3 \end{pmatrix}. \tag{14}$$

Now consider a string formed by the winding of $c^5$. The inner core behaviour is found by expanding the metric in the limit $\tau_5 \to \infty$, $\tau_4 =$ fixed, giving a diagonal metric with $G_{55} \simeq 3/\tau_5^2$. As this is the same $1/\rho_2^2$ scaling as for $T^2$, and by definition the potential is unimportant in the inner core, the inner core behaviour of $m^5 \equiv c^5 + i\tau^5$ derived from eq.(11) is identical to eq.(8) except the decompactification is to 8d! In Region III the metric is effectively frozen at its asymptotic value, so one again finds a $1/r^2$ dependence in the appropriate combination of moduli (upon diagonalising the asymptotic metric) as in eq.(7), and a tension of the same form as eq.(10). Only for the detailed outer-core matching does the full kinetic metric and potential need to be kept. Finally, as emphasised in Sec.4, in the deep inner core the fields are sourced by an effective 'magnetic' string in the decompactified 8d theory. In the case of an ALP from the RR $C_4$ this string in 8d arises from a $D3$ brane wrapped over the dual 2-cycle to $\tau_5$ (see also [48, 49]).

In all cases we have examined we have found qualitatively similar behaviour to that of Secs.3,4, though we do not know if this covers all axiverse string possibilities.

# 6 Phenomenology

We now briefly discuss aspects of the phenomenology. In the full theory axion strings always bound domain walls due to the tiny non-perturbatively generated mass of the ALP itself. The string/domain-wall network thus ultimately decays (for this reason axion strings were excluded from consideration in [54]). Nevertheless the presence of the decaying network can lead to important physical effects, such as gravitational waves [31–34]. In particular, as some of the axiverse ALP masses can be as small as $10^{-20}$ eV (for dark matter) or even $H_0 \sim 10^{-33}$ eV [19] (implementing quintessence), the network can survive to the post-recombination epoch or even the present. Moreover, if the axion couples to electromagnetism strings can give rise to quantised polarisation rotation of CMBR photons, as well as zero modes mandated by index theorems and thus associated forms of superconductivity [55–65].

As striking, the long-distance $1/r^2$ variation, eq.(7), of the associated non-ALP modulus $\text{Im}(m^i)$ leads to new effects characteristic of axiverse strings. One variation that is essentially guaranteed is the mass of the ALP itself, as almost always the non-perturbative effects that set the potential $|\tilde{V}| \sim \exp(-S[\text{Im}(m^i)])$ depend exponentially on $\text{Im}(m^i)$; here $S$ is a suitable instanton action. Moreover SM Yukawa couplings of the light quarks and leptons in string theory also often depend exponentially on moduli as they too are generated non-perturbatively [66–68], so masses and mixings of the light SM fermions can vary strongly in the vicinity of the string. These changes are typically equivalence principle violating.

Finally, we believe that our axiverse cosmic strings will be formed in the early universe by similar physics to that of the D-brane inflation production mechanisms of macroscopic cosmic F- and D-strings [54, 69–71].

These issues will be elucidated in future work.

# Acknowledgements

We are grateful to Ed Hardy for comments. HT thanks the STFC for a postgraduate studentship during the completion of this work.

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
