# Peer review of "On the Nature of Axiverse-Axion Cosmic Strings"

_SciPost Physics_

## Round 1 · Referee Report · Anonymous (Referee 1) · 2023-2-21

Strengths

1 - The paper is concise;
2 - The paper addresses compelling questions regarding axion physics.

Weaknesses

1 - The generality of the setting examined is not clear - in particular, it is not clear how the models under scrutiny relate to concrete string theory-originated effective field theories;
2 - Potential new results are not highlighted or worked out concretely, and the related sections are not expanded enough;
3 - As a consequence of the points above, it is then not clear how this work expands on pre-existing results in the literature.

Report

Axions can be helpful to efficiently address compelling physics problems, ranging from the strong CP problems to inflation.
The existence of axions within effective field theories can be justified by the UV completion thereof. Indeed, effective field theories originating from string theory naturally come with a copious amount of axions, besides other non-axionic fields.
The present manuscript revolves around the exploration of some phenomenological aspects of the effective field theories that include axions: namely, the presence of some solitonic solutions named ‘cosmic string solutions’.

Cosmic string solutions have been extensively studied in the last decades, and from various perspectives: from the earlier works in cosmology in the ‘80s to field theoretical or string theory-inspired models.
This work, after recalling some of the earlier works, focuses on cosmic strings that emerge in some effective field theories that should be inspired by those emerging from string theory.
In particular, it is noticed that the presence of a scalar potential for the scalar fields of the theory renders it non-trivial to find concrete cosmic string solutions.
The work further comments on possible phenomenological implications of the presence of the cosmic string solutions in such contexts.

Although the work is interesting, there are several criticalities that ought to be addressed.
For instance, it is unclear what is the level of generality of the analysis carried out in this work, as well as the relation of the models with concrete string theory effective field theory.
For instance, is equation (2) in Section 3 general? Or, does Section 5 contain “general” models?
The effective field theories examined in said sections seem rather particular.
For instance, the shape of the scalar potential (3) is only a particular case of the more general scalar potential that emerges, for instance, in 4D F-theory compactifications (see, for instance, the work by Grimm et al. in 2019).

It is also worth mentioning that the emergence and relevance of cosmic string solutions in string theory effective field theories have been studied since - for instance - the works by Greene et al. or Dabholkar et al. in 1990.
Moreover, recently, the study of cosmic string solutions (or, more general solitonic solutions) has become valuable for the so-called Swampland program.
Several new results have been discovered about cosmic string solutions in the past years within this program.
In particular, cosmic string solutions have been pointed out to be valuable for testing several conjectures of the program by Reece (arXiv:hep-th/1808.09966),
Moreover, the more recent work by Lanza et al. (arXiv:hep-th/2104.05726) considers cosmic string solutions in a huge variety of concrete string theory effective field theories, and discusses their regimes of validity in great detail.
Unfortunately, the current manuscript seems to ignore all these recent developments.
Indeed, some considerations and observations that are portrayed in this work were already contained - possibly examined in greater detail - in these earlier works.

On a more general level, it is not clear how the effective field theories examined, and the solutions obtained from those, rely on supersymmetry. In particular, is the supersymmetry preserved by the cosmic string solution?
Again, it is worth noticing that, for supersymmetric effective field theories, cosmic string solutions have been extensively studied and known since the works by Greene et al. and Dabholkar et al. in 1990.
And there are works in which cosmic string solutions with a non-trivial scalar potential are found - such as the work of Dvali et al. on ‘D-term strings’ (arXiv:hep-th/0312005).

Finally, the last section contains interesting considerations about possible phenomenological implications of the discussion put forward in the previous sections.
However, I feel that it could have been better expanded, and related to results in the existing literature.

In sum, I would suggest the publication of the manuscript only if substantial changes and improvements are performed.
In particular, the authors should stress what is actually new in this manuscript in relation to the results already achieved in literature that are momentarily ignored; moreover, in order to make the paper impactful for the community, I would suggest to further elaborate the qualitative arguments provided in the text: in particular, regarding the role of scalar potential for the cosmic string solutions (that is here not presented with concrete computations) and the phenomenological observations made in Section 6.

Requested changes

1 - Highlight new results in the current version of the manuscript;
2 - Better relate the discussion (such as the one in Section 5) with pre-existing results in the literature, stressing what was already found in the literature;
3 - Expand - in particular - Section 6 with concrete computations to support the qualitative statements made therein

---

## Round 1 · Referee Report · Anonymous (Referee 2) · 2023-3-4

Strengths

This paper examines axionic cosmic strings in the context of string theory axions. Although it is widely appreciated that many axions can arise in string theory, in phenomenological studies the differences in properties of these axions from conventional (e.g., KSVZ or DFSZ) axions are often neglected. At the same time, experimental efforts to detect axions and studies of their possible signals in astrophysics are becoming a significant subfield of particle physics. Thus, the topic is important and timely.

Weaknesses

The paper seems to make some strong assumptions, and doesn't engage much with recent related literature. (Details in the Report.)

Report

The authors emphasize that axiverse cosmic strings have cores that probe boundaries of moduli space, unlike conventional 4d field theory strings which simply have the PQ-breaking field going to zero in the core. This is an important and under-appreciated point. However, it has been emphasized in several other papers in recent years, which I suggest the authors read and consider citing in this context. First and foremost, Dolan et al. (arXiv:1701.05572) studied cosmic strings in the context of the idea that gravitational theories screen super-Planckian field ranges. They explicitly studied theories where the core of a string probes an infinite-distance limit of a radion field. This work is very closely related to the concepts contained in the current paper. Later, Reece (arXiv:1808.09966) emphasized that axionic string cores could probe infinite-distance limits in the context of some other Swampland conjectures.

More recently, papers by Lanza et al. which are cited (but only in a later section) as refs 48 and 49 undertook a comprehensive study of so-called "EFT strings" in the string landscape. These are strings whose tension goes to zero in a limit of moduli space. (This criterion is not the same as the infinite-distance limit mentioned above; the simplest examples obey both criteria, but some strings arising from D-branes wrapped on small cycles have an infinite-distance limit in their core but are not EFT strings in the Lanza et al. sense.) A more recent study involving this notion of EFT string is arXiv:2208.00009 by Cota et al., but perhaps this is less relevant for the current authors.

The authors remark (e.g. in the third paragraph of section 2) that moduli potentials are generated by non-perturbative effects, and that the "saxion" fields t^i can receive mass while the axion fields do not. Both of these statements are true, but they are in some tension with each other. Stabilization of saxion fields by non-perturbative superpotential effects tends to stabilize axions at a similar mass scale. If one wants to parametrically separate the axions from the saxions, one can rely on supersymmetry-breaking stabilization by Kähler potential terms, which leave the axion untouched--but which are perturbative effects, not non-perturbative ones. This is nicely explained in a paper by Conlon, hep-th/0602233. (This paper also discusses the scenario for which the slightly paper of Svrcek and Witten is cited in the second paragraph of section 4.) Because of this, I am not sure if equation (3) is a plausible form for a potential for a saxion in a case where the axion is parametrically lighter.

In general, the paper emphasizes logarithmic kinetic terms for moduli. This is generally valid for moduli in asymptotic regions of moduli space. But this is not necessarily the most relevant case for all axions. For instance, axions arising on a small cycle within a large-volume compactification can have axionic strings arising from D-branes wrapped on small cycles, which remain of fixed size in asymptotic weak-coupling limits. (Notice that one cannot send tau_4 to infinity in the example discussed around equation (14), because it measures the size of a hole in the space; it must be smaller than tau_5.) Correspondingly, the comment below equation (12) that the metric scales as 1/t^2 may not be relevant for such moduli. The properties of the corresponding axionic strings may differ from those that are mostly discussed in this paper.

I'm also confused by the comment below equation (14) that this scenario has a decompactification to 8d. The overall volume scales as tau_5^{3/2}, so the limit of large tau_5 actually makes all six dimensions large (not just those of the 4-cycle) and represents a decompactification to 10d. I am not sure if the comment is supposed to imply some less obvious scaling limit.

Section 6 remarks that Yukawa couplings in string theory are generated non-perturbatively. I'm not sure if this was intended as a generic statement or one about specific models; for instance, it is true of intersecting D6-brane models in Type IIA, but not of intersecting D7-brane models in Type IIB, where the Yukawa couplings arise perturbatively.

The statement that axiverse strings will be produced in the early universe is an interesting one, which is not obvious to me. I hope to see more details in the promised future work.

Requested changes

  1. Refer to the existing literature more, and explain how this work fits in the context of studies in the last several years.

  2. Better justify the form of the potential in equation (3).

  3. Comment on the extent to which the conclusions are generic or apply to, for instance, mostly the case of branes wrapped on large-volume cycles.

  4. Address other comments above, e.g., about the 8d decompactification limit or the non-perturbative nature of Yukawa couplings.

---

## Editorial Decision

awaiting_resubmission